# Clustering long-term health conditions among 67728 people with multimorbidity using electronic health records in Scotland

**Adeniyi Francis Fagbamigbe**[1,2,3,4]*, **Utkarsh Agrawal**[5], **Amaya Azcoaga-Lorenzo**[1,6], **Briana MacKerron**[1], **Eda Bilici Özyiğit**[7], **Daniel C. Alexander**[7], **Ashley Akbari**[8], **Rhiannon K. Owen**[8], **Jane Lyons**[8], **Ronan A. Lyons**[8], **Spiros Denaxas**[9,10], **Paul Kirk**[11], **Ana Corina Miller**[12], **Gill Harper**[13], **Carol Dezateux**[13], **Anthony Brookes**[14], **Sylvia Richardson**[11], **Krishnarajah Nirantharakumar**[15], **Bruce Guthrie**[16], **Lloyd Hughes**[1], **Umesh T. Kadam**[17], **Kamlesh Khunti**[18], **Keith R. Abrams**[19], **Colin McCowan**[1]

1 School of Medicine, University of St Andrews, St Andrews, United Kingdom, 2 Department of Epidemiology and Medical Statistics, University of Ibadan, Ibadan, Nigeria, 3 Institute of Applied Health Sciences, University of Aberdeen, Aberdeen, United Kingdom, 4 Research Methods and Evaluation Unit, Institute for Health & Wellbeing, Coventry University, Coventry, United Kingdom, 5 Nuffield Department of Primary Care Health Science, University of Oxford, Oxford, United Kingdom, 6 Hospital Rey Juan Carlos, Instituto de Investigación Sanitaria Fundación Jimenez Diaz, Madrid, Spain, 7 Centre for Medical Image Computing, Department of Computer Science, UCL, London, United Kingdom, 8 Population Data Science, Swansea University Medical School, Swansea University, Swansea, United Kingdom, 9 Institute of Health Informatics, UCL, London, United Kingdom, 10 British Heart Foundation Data Science Centre, London, United Kingdom, 11 MRC Biostatistics Unit, University of Cambridge, Cambridge, United Kingdom, 12 Centre for Public Health, Institute of Clinical Science, Queen's University Belfast, Belfast, United Kingdom, 13 Clinical Effectiveness Group, Wolfson Institute of Population Health, Queen Mary University of London, London, United Kingdom, 14 Department of Genetics & Genome Biology, University of Leicester, Leicester, United Kingdom, 15 Public Health, University of Birmingham, Birmingham, United Kingdom, 16 Advanced Care Research Centre, Usher Institute, University of Edinburgh, Edinburgh, United Kingdom, 17 Department of Population Health Sciences, University of Leicester, Leicester, United Kingdom, 18 Diabetes Research Centre, University of Leicester, Leicester, United Kingdom, 19 Department of Statistics, University of Warwick, Coventry, United Kingdom

* aff5@st-andrews.ac.uk, franstel74@yahoo.com

**Data Availability Statement:** A data dictionary covering the data sources used in this study and the analysis codes are deposited at https://github.

## Abstract

There is still limited understanding of how chronic conditions co-occur in patients with multimorbidity and what are the consequences for patients and the health care system. Most reported clusters of conditions have not considered the demographic characteristics of these patients during the clustering process. The study used data for all registered patients that were resident in Fife or Tayside, Scotland and aged 25 years or more on 1st January 2000 and who were followed up until 31st December 2018. We used linked demographic information, and secondary care electronic health records from 1st January 2000. Individuals with at least two of the 31 Elixhauser Comorbidity Index conditions were identified as having multimorbidity. Market basket analysis was used to cluster the conditions for the whole population and then repeatedly stratified by age, sex and deprivation. 318,235 individuals were included in the analysis, with 67,728 (21·3%) having multimorbidity. We identified five distinct clusters of conditions in the population with multimorbidity: alcohol misuse, cancer, obesity, renal failure, and heart failure. Clusters of long-term conditions differed by age, sex and socioeconomic deprivation, with some clusters not present for specific strata and

com/fadeniyi123/MM_HDRUK. The data used in this study are sensitive and are not publicly available. Access to the data is by application to the Health Informatics Centre, University of Dundee, Scotland (hicsupport@dundee.ac.uk) using their standard governance and access processes.

**Funding:** CMC: This work was supported by Health Data Research UK (HDR UK) Measuring and Understanding Multimorbidity using Routine Data in the UK (HDR-9006; CFC0110). Health Data Research UK (HDR-9006) is funded by: UK Medical Research Council, Engineering and Physical Sciences Research Council, Economic and Social Research Council, the National Institute for Health Research (England), Chief Scientist Office of the Scottish Government Health and Social Care Directorates, Health and Social Care Research and Development Division (Welsh Government), Public Health Agency (Northern Ireland), British Heart Foundation, and Wellcome Trust. The funders had no role in study design, data collection and analysis, decision to publish, or preparation of the manuscript. NO

**Competing interests:** The authors have declared that no competing interests exist.

others including additional conditions. These findings highlight the importance of considering demographic factors during both clustering analysis and intervention planning for individuals with multiple long-term conditions. By taking these factors into account, the healthcare system may be better equipped to develop tailored interventions that address the needs of complex patients.

## Background

Multimorbidity, also known as multiple long-term conditions, is the co-existence of two or more long-term conditions within an individual [1]. It is now the norm in ageing populations, with this group of patients being inherently heterogeneous [2, 3]. The estimated prevalence of multimorbidity varies considerably depending on the population studied, the specific list conditions that are included in the analysis and the data sources used., the [4], but consistent findings show that multimorbidity is common, more frequent in older people, women, and socioeconomically deprived populations [5, 6] The relationship between socioeconomic deprivation and multimorbidity is complex [7, 8], but there is evidence that the less affluent have earlier onset and more rapid accumulation of conditions resulting in widening inequalities into old age [9].

While most clinical guidelines focus on managing individual conditions, the number of individuals with multimorbidity is increasing, causing major concerns for the delivery of care in an already constrained healthcare system with competing needs [10, 11]. Prioritising interventions for high-risk groups is vital as healthcare systems strive toward the sustainability of service delivery. There is considerable evidence suggesting that the current disease-based approach to managing individuals with multimorbidity is associated with a variety of poor outcomes, including inadequate preventative care and access to rehabilitation services [12], repeated referrals for specialist care [13] and increased healthcare costs [14].

Understanding how conditions cluster is a key element in unravelling determinants and the delivery of future healthcare. Little is known about how disease clusters contribute to multimorbidity and complex multimorbidity (defined as having 4 or more multiple long-term conditions) [3] across age, sex, and socioeconomic deprivation of individuals [15]. Most studies that report clusters of conditions do so without considering the demographic characteristics of the patients which could affect the nature of observed clusters [16, 17]. Some approaches to clustering within the multimorbidity literature, aim to classify patients based on their conditions and place them into similar groups while other approaches aim to identify groups of conditions which are present in individuals more frequently than expected.

Recent work by Kuan et al. showed variability in the most common single conditions across the life course and also by sex [18]. Other studies have reported that socioeconomic status also impacts the development of multimorbidity during different periods [19]. A deeper understanding of how the different demographic characteristics associated with and contributing to clusters of conditions in patients with multimorbidity is needed to help clinicians in the management of those patients and to prepare the health systems to provide adequate management for these complex patients. This study aims to assess the prevalence of multimorbidity and complex multimorbidity by age, sex and area-level socioeconomic deprivation. We also identified the most common condition among patients with multimorbidity, and key clusters of disease, stratified by age, sex and socioeconomic deprivation.

## Methods

### Study design and population

The population for this study were residents of Fife and Tayside, Scotland who were aged at least 25 years old on 1st January 2000 and alive on 31 December 2018, when a cross-sectional analysis of all live patients was performed. The data was generated for a study exploring multi-morbidity across different countries in the UK. Exploration of the data showed a strong socio-economic and age gradient in terms of specific individual conditions and prevalence multimorbidity [20] which we felt warranted further exploration [20]. We ascertained the dates of death from National Record Scotland death certificates and the population register. The dataset used linked pseudonymised health and demographic data held by the Health Informatics Centre (HIC) at the University of Dundee.

Health care in Scotland is provided free at the point of care by the taxpayer-funded National Health Service (NHS). NHS Tayside and Fife are two separate Health Boards that provide specialist and secondary care services and contract with general practices that provide primary medical care to an approximate population of 800,000 individuals. The unique Community Health Index (CHI) number allocated to individuals at the point of registration with a general practice (GP), is used across the NHS. Demographic information, hospital admission and day-case records, cancer registry and mental health inpatient were linked to the death data from 1st January 2000 and Emergency Department (ED) attendances from 1st January 2017 onwards.

### Multimorbidity definitions

All hospital admissions, psychiatric hospital admissions, outpatients, cancer registry and emergency department records over the period were examined, and all the International Classification of Diseases (ICD)-10 codes were extracted. We identified 31 conditions listed in the Elixhauser Comorbidity Index [21] (see S1 Table) based on the presence of ICD-10 codes relevant to individual conditions for the presence of the condition and the first record of diagnosis date during the study period. Depression and psychoses are examples of mental health long-term conditions included within the Elixhauser index whilst weight loss and cancer are some of the physical conditions listed (see S1 Table for a full list of conditions and related codes). The Elixhauser Index was chosen as previous reviews have suggested it is well established for use with electronic health records using ICD10 codes [22]. All individuals with two or more conditions were defined as having multimorbidity, and those with four or more were identified as having complex multimorbidity.

### Explanatory variables

The patient's age was calculated on 31st December 2018 and grouped as 44–49, 50–59, 60–69, 70–79, and 80+ years. Sex was recorded as male/female, and socioeconomic status was measured by the quintiles of the Scottish Index of Multiple Deprivation (SIMD), a postcode-assigned measure of small area (data zone) deprivation. SIMD uses seven domains (income, employment, health, education, housing, crime, access to services) to score data zones in different aspects of deprivation and is then ranked and grouped into quintiles [23].

### Data management and statistical analysis

Frequencies and proportions of individuals with the conditions and the prevalence of multi-morbidity and complex multimorbidity within each stratum (age, sex, deprivation) were reported and relevant tests for differences were used. Analysis of clusters was performed in

three phases: (i) the whole population with multimorbidity, (ii) by age, sex and socioeconomic deprivation, and (iii) by their interactions. The rate of co-occurrence between pairs of the long-term conditions is presented in S1 Fig. To allow for the clustering of conditions across the characteristics of the population, the market basket analysis (MBA—also known as association rule mining) using the Apriori algorithm was used [24, 25]. The dissimilarity command with the Jaccard option within the MBA was used to identify clusters among the conditions stratified by age, sex and SIMD [24, 25]. A cluster is a group of conditions with shorter distances to themselves than to other conditions in a binary matrix of conditions. The dissimilarity function organises the conditions by cluster so that the conditions within clusters are closer together than those in different clusters, and therefore more likely to co-occur. We used MBA because it has been reported as more efficient for binary (present/absent) outcomes than the hierarchical cluster analysis that was originally built for quantitative outcomes [24, 25]. It also allows an individual to "belong" to more than one cluster if they have a large number of different conditions. The method computes and returns distances for binary data in transactions which can be used for grouping and clustering [24]. The optimal number of clusters from the dissimilarity clustering was determined using the Elbow method [26]. For clustering, we considered only conditions that had at least 5% prevalence in the population with multimorbidity (see the S2 Table). The clusters are summarised in Tables 2 and 3 and S3–S5 Tables, and the generated dendrograms are presented in S2–S4 Figs. R and Stata version 17 were used for the analysis.

### Ethical approval

HIC provided a linked dataset within a Safe Haven environment for this study. The dataset was obtained under HIC Standard Operating Procedures (SOP). NHS Tayside Research Ethics Committee have approved these SOPs (18/ES/0126). The School of Medicine Ethics Committee, acting on behalf of the University of St Andrews Teaching and Research Ethics Committee approved the project (UTREC MD15619 approved 30th June 2021). As the study data are de-identified, consent from individual patients was not required.

### Results

Overall, 318,235 people aged 44 years and over were included in the analysis, with 67,728 (21·3%) identified as having multimorbidity (2+ conditions), while 20,123(6·3%) were also classed as having complex multimorbidity (4+ conditions). The mean (SD) age of the people with multimorbidity was 72·8(7·1) years, 31439(46·4%) were men, 13955(20·6%) were most deprived and 12268(18·1%) were least deprived. The prevalence of both multimorbidity and complex multimorbidity in the whole population was similar for both sexes and increased significantly with age (Fig 1) and with increasing socioeconomic deprivation. more women in younger age groups (44–59) have multimorbidity compared to men, whereas in individuals aged 60 and above, men have a higher prevalence of multimorbidity (Table 1).

The most common condition among all patients with multimorbidity was uncomplicated hypertension (53%) followed by chronic pulmonary disease (27%) while the least common was AIDS/HIV (0·1%) (see S2 Table). The most frequent conditions among 44-49-year-olds with multimorbidity were chronic pulmonary disease (29%), alcohol abuse (28%), and depression (28%), compared with uncomplicated hypertension (65%), cardiac arrhythmia (37%) and solid tumour without metastasis (34%) among those aged 80+ years. Uncomplicated hypertension is the leading condition among both the most deprived (50%) and least deprived (55%) patients with multimorbidity (see S3 Table). The rate of co-occurrence of pairs of conditions per 1000 patients with multimorbidity is shown in S1 Fig. The commonest pairs of conditions

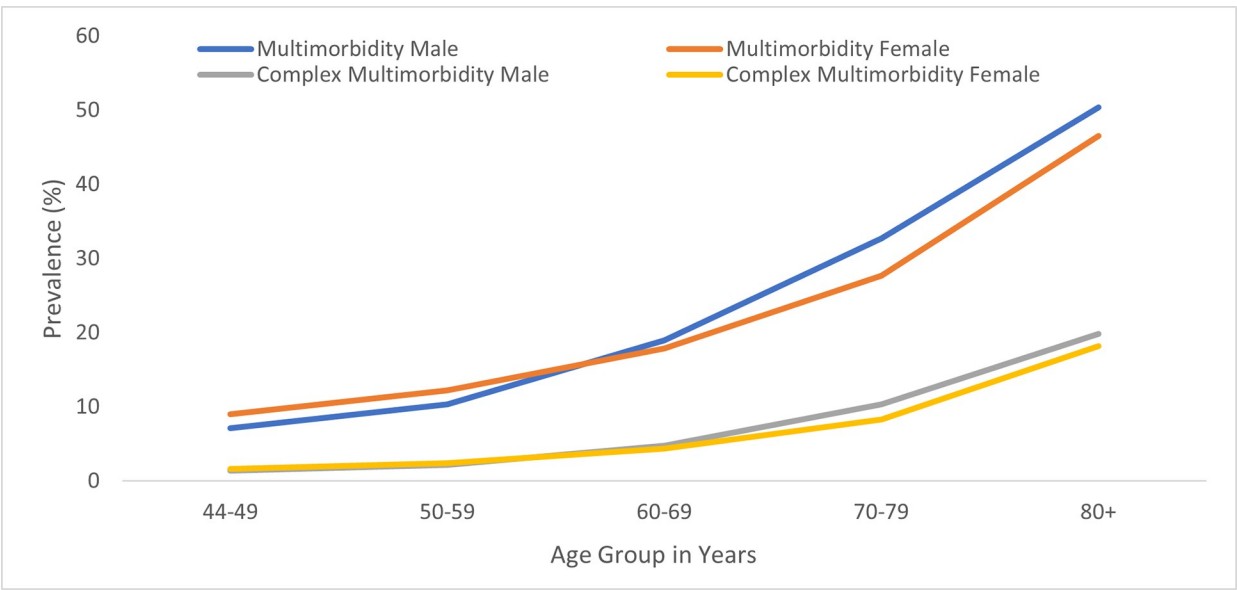

**Fig 1. Prevalence of multimorbidity by sex and age.**

were uncomplicated hypertension and uncomplicated diabetes (32·5 per 1000 patients with multimorbidity), uncomplicated hypertension and cardiac arrhythmia (30·0/1000), chronic pulmonary disease and uncomplicated hypertension (28·1/1000) and uncomplicated hypertension / solid tumour without metastasis (26·6/1000).

## Clustering of conditions among people with multimorbidity

**All people with multimorbidity.**   In the total population with multimorbidity, conditions were grouped into five clusters (Table 2).

Cluster 1: Alcohol misuse ClusterIn cluster 1, twenty-six percent (n = 17366) of the people with multimorbidity had at least two of alcohol abuse, other neurological disorders, and depression., The mean age of people in cluster 1 was 66·4 (standard deviation = 12·5) years, 8891(51%) of them were women, 4098(24%) and 2049(12%) from the most deprived and least deprived groups respectively (S4A Table).

Cluster 2: Cancer Cluster

10766(56%) of the 19,123 people with at least two solid tumours without metastasis, and metastatic cancer were women with 2945(16%) and 3748(19%) from the most deprived and least deprived groups respectively (S4A Table).

Cluster 3: Obesity Cluster

Obesity, uncomplicated hypertension, chronic pulmonary disease, rheumatoid arthritis/ collagen disorders, hypothyroidism, and uncomplicated diabetes formed cluster 3, 55105 (81·4%) of the people with multimorbidity have at least two of the conditions in cluster 3.

Cluster 4: Renal Failure Cluster

The conditions in cluster 4 are peripheral vascular disorders, renal failure, fluid & electrolyte disorders, and deficiency anaemia.

Cluster 5: Heart Failure Cluster

The conditions that formed cluster 5 are pulmonary circulation disorders, valvular disease, congestive heart failure and cardiac arrhythmias.

**Table 1. Distribution of population by background characteristics and level of multimorbidity.**

| Characteristics | All individuals | Distribution of multimorbidity patients (n, %) | Prevalence of multimorbidity (2+) (n, %) | Prevalence of complex multimorbidity (4+) (n, %) |
|---|---|---|---|---|
| **All** | 318,235 | 67728 (100) | 67,728 (21·3) | 20,123 (6·3) |
| **Sex** | | | | |
| F | 169,086 | 36289 (53·6) | 36,289 (21·5) | 10,711 (6·3) |
| M | 149,149 | 31439 (46·4) | 31,439 (21·1) | 9,412 (6·3) |
| **Age (years) Mean (SD)** | 64.6(6.9) | | 72·8(7·1) | 75.3(6.8) |
| 44–49 | 40,023 | 3232 (4·8) | 3,232 (8·1) | 595 (1·5) |
| 50–59 | 90,719 | 10226 (15·1) | 10,226 (11·3) | 2,088 (2·3) |
| 60–69 | 80,486 | 14827 (21·9) | 14,827 (18·4) | 3,687 (4·6) |
| 70–79 | 66,379 | 19924 (29·4) | 19,924 (30·0) | 6,098 (9·2) |
| 80+ | 40,628 | 19519 (28·8) | 19,519 (48·0) | 7,655 (18·8) |
| **Deprivation quintile** | | | | |
| Most deprived 1 | 52,958 | 13955 (20·6) | 13,955 (26·4) | 4,449 (8·4) |
| 2 | 58,468 | 14242 (21·1) | 14,242 (24·4) | 4,458 (7·6) |
| 3 | 62,535 | 13105 (19·3) | 13,105 (21·0) | 3,879 (6·2) |
| 4 | 64,568 | 12271 (18·1) | 12,271 (19·0) | 3,485 (5·4) |
| Least deprived 5 | 69,416 | 12268 (18·1) | 12,268 (17·7) | 3,259 (4·7) |
| Missing | 10,290 | 1887 (2·8) | 1,887 (18·3) | 593 (5·8) |
| **Women /Age (years)** | | | | |
| 44–49 | 20,498 | 1836 (5·1) | 1,836 (9·0) | 324 (1·6) |
| 50–59 | 46,849 | 5723 (15·8) | 5,723 (12·2) | 1,137 (2·4) |
| 60–69 | 41,586 | 7440 (20·4) | 7,440 (17·9) | 1,829 (4·4) |
| 70–79 | 35,497 | 9823 (27·1) | 9,823 (27·7) | 2,930 (8·3) |
| 80+ | 24,656 | 11467 (31·6) | 11,467 (46·5) | 4,491 (18·2) |
| **Men/Age (years)** | | | | |
| 44–49 | 19,525 | 1396 (4·4) | 1,396 (7·1) | 271 (1·4) |
| 50–59 | 43,870 | 4503 (14·3) | 4,503 (10·3) | 951 (2·2) |
| 60–69 | 38,900 | 7387 (23·5) | 7,387 (19·0) | 1,858 (4·8) |
| 70–79 | 30,882 | 10101 (32·2) | 10,101 (32·7) | 3,168 (10·3) |
| 80+ | 15,972 | 8052 (25·6) | 8,052 (50·4) | 3,164 (19·8) |
| **Women/ Deprivation** | | | | |
| Most deprived | 27,874 | 7516 (20·7) | 7,516 (27·0) | 2,396 (8·6) |
| 2 | 31,132 | 7777 (21·4) | 7,777 (25·0) | 2,445 (7·9) |
| 3 | 33,201 | 7040 (19·4) | 7,040 (21·2) | 2,055 (6·2) |
| 4 | 34,341 | 6461 (17·8) | 6,461 (18·8) | 1,802 (5·2) |
| Least deprived | 37,128 | 6413 (17·7) | 6,413 (17·3) | 1,664 (4·5) |
| Missing | 5,410 | 1082 (3·0) | 1,082 (20·0) | 349 (6·5) |
| **Men/Deprivation** | | | | |
| Most deprived | 25,084 | 6439 (20·4) | 6,439 (25·7) | 2,053 (8·2) |
| 2 | 27,336 | 6465 (20·6) | 6,465 (23·7) | 2,013 (7·4) |
| 3 | 29,334 | 6065 (19·3) | 6,065 (20·7) | 1,824 (6·2) |
| 4 | 30,227 | 5810 (18·5) | 5,810 (19·2) | 1,683 (5·6) |
| Least deprived | 32,288 | 5855 (18·6) | 5,855 (18·1) | 1,595 (4·9) |
| Missing | 4,880 | 805 (2·6) | 805 (16·5) | 244 (5·0) |

**Table 2. Multimorbidity clusters of the conditions among the whole population with multimorbidity, sex and deprivation subgroups.**

| Cluster+ | Whole population | Males | Females | Most deprived | Least deprived |
|---|---|---|---|---|---|
| 1 Alcohol misuse Cluster | Alcohol Abuse Other Neurological Disorders Depression | Alcohol abuse Other Neurological Disorders Depression Liver Disease | Alcohol Abuse Other Neurological Disorders Depression | Alcohol abuse Other Neurological Disorders Depression Liver Disease Drug Abuse | Alcohol abuse Other Neurological Disorders Depression |
| 2 Cancer Cluster | Solid Tumour w/o Metastasis Metastatic Cancer | Solid Tumour w/o Metastasis Metastatic Cancer | Solid Tumour w/o Metastasis Metastatic Cancer | Solid Tumour w/o Metastasis Metastatic Cancer | Solid Tumour w/o Metastasis Metastatic Cancer |
| 3 Obesity Cluster | Obesity Chronic Pulmonary Disease Uncomplicated Hypertension Uncomplicated Diabetes Rheumatoid Arthritis/Collagen Hypothyroidism | Obesity Chronic Pulmonary Disease Uncomplicated Hypertension Uncomplicated Diabetes Rheumatoid Arthritis/Collagen | Obesity Chronic Pulmonary Disease Uncomplicated Hypertension Uncomplicated Diabetes Rheumatoid Arthritis/Collagen Hypothyroidism | Obesity Chronic Pulmonary Disease Uncomplicated Hypertension Uncomplicated Diabetes | Obesity Chronic Pulmonary Disease Cardiac Arrhythmia Uncomplicated Hypertension Uncomplicated Diabetes Rheumatoid Arthritis/Collagen Hypothyroidism |
| 4 Renal Failure Cluster | Peripheral Vascular Disorders Renal Failure Fluid & Electrolyte Disorders Deficiency Anaemia | Peripheral Vascular Disorders Renal Failure Fluid & Electrolyte Disorders | Peripheral Vascular Disorders Renal Failure Fluid & Electrolyte Disorders Deficiency Anaemia | Peripheral Vascular Disorders Renal Failure Fluid & Electrolyte Disorders Deficiency Anaemia Hypothyroidism Rheumatoid Arthritis/Collagen | Peripheral Vascular Disorders Renal Failure Fluid & Electrolyte Disorders Deficiency Anaemia |
| 5 Heart Failure Cluster | Valvular Disease Congestive Heart Failure Cardiac Arrhythmia Pulmonary Circulation Disorders | Valvular Disease Congestive Heart Failure Cardiac Arrhythmia | Valvular Disease Congestive Heart Failure Cardiac Arrhythmia Pulmonary Circulation Disorders | Valvular Disease Congestive Heart Failure Cardiac Arrhythmia Pulmonary Circulation Disorders | Valvular Disease Congestive Heart Failure Pulmonary Circulation Disorders |

*only conditions with at least 5% prevalence within each specific population subgroup were clustered +An individual can "belong" to more than one cluster.

In all, the percentages of people with multimorbidity from the most deprived groups were higher than the people from the least deprived group except for cluster 2. The clusters of conditions identified for strata of sex, age and socio-economic deprivation are presented in (Table 3) and in S4, S5 and S6 Tables and S2, S3 and S4 Figs.

Looking at stratification by sex and social deprivation, the identified clusters had a core set of conditions across strata. "The core conditions included alcohol misuse, other neurological disorders and depression in the alcohol misuse cluster and solid tumour without metastasis and metastatic cancer in the cancer cluster". However, some clusters for specific strata also have additional conditions within the clusters. For instance, most deprived people had additional conditions such as drug abuse in the alcohol misuse cluster (See Table 2). There are similarities in the number of clusters formed among these conditions across sex and deprivation quintiles. Identifying clusters for the different age groups, conditions in cluster 1 among the youngest (44–49 years) and those in their 50s are similar while cluster 1 in the 60s and 70s are similar but such cluster did not exist in those aged 80 years or older. Cluster 2 in the youngest group (44–49 years), and Cluster 3 in the 50s and 60s look similar while there were additional conditions as the patients grew older (Table 3).

For the most deprived populations aged 80 years and over, drug abuse, alcohol abuse, psychosis, and depression formed a cluster which affected 1 in 5 people with multimorbidity in

**Table 3. Multimorbidity clusters of the conditions by age*,^@.**

| Clusters*,^@+ | Whole population | 44–49 years | 50–59 years | 60–69 years | 70–79 years | 80+ years |
|---|---|---|---|---|---|---|
| Alcohol misuse cluster | Alcohol Abuse Other Neurological Disorders Depression | Alcohol abuse Other Neurological Disorders Depression Psychoses Drug Abuse Liver Disease | Alcohol abuse Depression Psychoses Drug Abuse | Alcohol abuse Other Neurological Disorders Depression Liver Disease | Alcohol abuse Other Neurological Disorders Depression | |
| Cancer cluster | Solid Tumour w/o Metastasis Metastatic Cancer | Solid Tumour w/o Metastasis Fluid & Electrolyte Disorders Cardiac Arrhythmia Rheumatoid Arthritis/ Collagen Deficiency Anaemia Hypothyroidism | Solid Tumour w/o Metastasis Metastatic Cancer | Solid Tumour w/o Metastasis Metastatic Cancer | Solid Tumour w/o Metastasis Metastatic Cancer | Solid Tumour w/o Metastasis Uncomplicated Hypertension Uncomplicated Diabetes Chronic Pulmonary Disease Cardiac Arrhythmia Renal Failure |
| Obesity cluster | Obesity Chronic Pulmonary Disease Uncomplicated Hypertension Uncomplicated Diabetes Rheumatoid Arthritis/Collagen Hypothyroidism | Obesity Uncomplicated Hypertension Uncomplicated Diabetes | Obesity Chronic Pulmonary Disease Uncomplicated Hypertension Uncomplicated Diabetes Rheumatoid Arthritis/Collagen Hypothyroidism | Obesity Chronic Pulmonary Disease Uncomplicated Hypertension Uncomplicated Diabetes Rheumatoid Arthritis/ Collagen Hypothyroidism | Chronic Pulmonary Disease Uncomplicated Hypertension Uncomplicated Diabetes Rheumatoid Arthritis/ Collagen | |
| Renal Failure cluster | Peripheral Vascular Disorders Renal Failure Fluid & Electrolyte Disorders Deficiency Anaemia | | Fluid & Electrolyte Disorders Deficiency Anaemia Liver Disease Other Neurological Disorders | Peripheral Vascular Disorders Renal Failure Fluid & Electrolyte Disorders Deficiency Anaemia | Peripheral Vascular Disorders Renal Failure Fluid & Electrolyte Disorders Deficiency Anaemia | Peripheral Vascular Disorders Pulmonary Circulation Disorders Congestive Heart Failure Valvular Disease |
| Heart Failure cluster | Valvular Disease Congestive Heart Failure Cardiac Arrhythmia Pulmonary Circulation Disorders | | Valvular Disease Congestive Heart Failure Cardiac Arrhythmia | Valvular Disease Congestive Heart Failure Cardiac Arrhythmia | Valvular Disease Congestive Heart Failure Cardiac Arrhythmia | |
| others | | | | | Obesity Hypothyroidism Pulmonary Circulation Disorders | Metastatic Cancer Other Neurological Disorders Fluid & Electrolyte Disorders Hypothyroidism Deficiency Anaemia Rheumatoid Arthritis/ Collagen |

*only conditions with at least 5% prevalence within each specific population subgroup were clustered ^efforts were made to align clusters that were similar across different populations. @Blank cells exist for certain age where the identified conditions are not present or clustered. +An individual can "belong" to more than one cluster.

this subgroup, but these conditions were not prominent among the older least deprived population. This would suggest that those planning initiatives aimed at different populations of people with multimorbidity should be aware that underlying clusters of disease will be different. Alcohol and drug abuse formed part of a cluster (liver disease, psychosis, alcohol abuse, drug, depression, chronic pulmonary disease and other neurological disorders) among 83% of the

**Table 4. Multimorbidity clusters of the conditions among the whole population with multimorbidity.**

| Population subgroup | Cluster | Conditions* | No of people in cluster n(%) | Mean age (std dev) | Women % | % in Most Deprived | % in the least Deprived |
|---|---|---|---|---|---|---|---|
| **Whole Population** N = 67728 | Alcohol misuse cluster | Alcohol Abuse Other Neurological Disorders Depression | 17366 (25·6) | 66·4 (12·5) | 8891 (51·2) | 4098 (23·6) | 2049 (11·8) |
| | Cancer cluster | Solid Tumour w/o Metastasis Metastatic Cancer | 19123 (28·2) | 74·7 (11·1) | 10766 (56·3) | 2945 (15·4) | 3748 (19·6) |
| | Obesity cluster | Obesity Chronic Pulmonary Disease Uncomplicated Hypertension Uncomplicated Diabetes Rheumatoid Arthritis/Collagen Hypothyroidism | 55105 (81·4) | 72·6 (12) | 29757 (54) | 10250 (18·6) | 8872 (16·1) |
| | Renal Failure cluster | Peripheral Vascular Disorders Renal Failure Fluid & Electrolyte Disorders Deficiency Anaemia | 20771 (30·7) | 75·8 (11·9) | 11736 (56·5) | 3884 (18·7) | 3199 (15·4) |
| | Heart Failure cluster | Valvular Disease Congestive Heart Failure Cardiac Arrhythmia Pulmonary Circulation Disorders | 23497 (34·7) | 75·7 (12) | 11020 (46·9) | 3854 (16·4) | 4229 (18·0) |

most deprived patients with multimorbidity aged 44–49 years. However, they contributed to a smaller cluster (psychosis, alcohol abuse, drug, depression) among only two-fifths of their least deprived counterparts.

About 26% of the patients with multimorbidity have at least two of the conditions in the alcohol misuse cluster, with a mean (standard deviation) age of 66.4(12.5) years, among which 51% were females, 24% and 12% from most deprived and least deprived groups respectively (Table 4). Fifty-six percent of the people with the conditions in the cancer cluster were females with 15% and 20% from the most deprived and least deprived groups respectively. About 8 of every 10 patients with multimorbidity have at least two of the conditions in the obesity cluster. The percentages of patients with multimorbidity from the most deprived groups were higher than the people from the least deprived group across all the clusters except for the cancer cluster.

## Discussion

In this population, where the presence of long-term conditions was ascertained using secondary care data, the prevalence of multimorbidity and complex multimorbidity was similar for both men and women and increased with age and socioeconomic deprivation. Several previous studies have identified clusters that include cardiovascular-metabolic conditions, mental health issues, and musculoskeletal disorders [27]. A recent study has also demonstrated variations in clusters by age, as well as differences in mortality and service utilisation [28]. In addition, our study reported how long-term conditions cluster differently based on age, sex, and socioeconomic deprivation.

The differences between the clusters of conditions by social deprivation and age support that the overall population with multimorbidity are essentially heterogeneous groups of patients with different conditions and hence different needs.

Clustering populations that were stratified by age and deprivation showed differences in people aged over 80 years, although there seemed less variability in people aged 44–49 years. These findings highlight that although the relationship between levels of deprivation age and

multimorbidity is well known [19, 29], there is much less known about differences in clusters of conditions for these characteristics [19].

The identified clusters strongly correspond to current medical knowledge, demonstrating well-known associations between conditions such as alcohol abuse and depression. Across the population, hypertension and cardiac arrhythmias were the study population's most prevalent pair of conditions, which supports the known relationship between hypertension and heart diseases [30]. Our analysis shows association, not causality but it may be possible to surmise the drivers of specific clusters as identified. Conditions most prevalent in our most deprived population groups include alcohol and drug misuse, depression and obesity which are all known to be associated with social factors. Other identified clusters are likely to have more physiological drivers e.g. hypertension through to heart diseases.

The choice of how to define multimorbidity is important in terms of conditions and risk factors. Obesity and hypertension can be considered as both conditions that require management and as risk factors contributing to the development of other health problems. Our findings suggest a high prevalence of obesity among individuals aged 44–49 years old with multimorbidity. This undoubtedly places a significant burden on both health and social care services, given the available evidence on how obesity can reduce life expectancy and healthy life expectancy [31]. The younger age group also had alcohol misuse as a key condition. This supports a recent report on alcohol-related harm with risk factors rooted foremost in socioeconomic determinants [32].

## Study strengths and limitations

The study population was drawn from two Scottish Health Boards with comprehensive health records over a long period. The use of well-defined conditions and ICD-10 codesets to identify each condition allows other researchers to explore multimorbidity using the same methods. Using market basket analysis to cluster conditions rather than classifying patients into mutually exclusive groups meant patients could be present in more than one cluster depending on the conditions they had Using the same approach across different strata meant comparisons were down to the underlying data rather than simply different populations using different methods.

The data used to identify conditions were hospital records from secondary care These will underestimate the occurrence of conditions as less severe cases might not have been captured. Some limitations of this work relate to the choice of the Elixhauser Index to identify underlying conditions. Some common conditions such as myocardial infarction (ICD10 code I21) are not among the 31 conditions identified in the Elixhauser Index and a few individual conditions may be a progression of a single condition, such as uncomplicated diabetes to diabetes with chronic complications. However, people with this progression of diseases also have other conditions and the clustering will be unaffected as only conditions with more than 5% prevalence were clustered. Each condition has a mutually exclusive code set meaning that different ICD10 codes are related only to one condition. We made an a priori decision to only study the 31 conditions listed and to treat them all as separate. Recent work from Ho et al. has suggested a more complete list of underlying individual conditions which may change the identified clusters [4] but is unlikely to change the fact that clusters will vary in different age groups, gender or socioeconomic groups.

Similarly, there have been several different methods used to identify clusters within multimorbid populations but our choice of Market Basket Analysis as a methodological tool is unlikely to be the cause of differences when examining strata. However, the clusters generated by market basket analysis are based on empirical patterns in the data and only show

associations between different conditions, it does not show any causal relationships between those conditions. We reported on clusters of multimorbidity in people alive on 31st December 2018, if we had included those who had died throughout the period, we may have seen some differences in the identified clusters. The measure of deprivation used in this study is allocated at a postcode level, but it is a small area approximation rather than a direct measure of individual deprivation.

The naming of the clusters was discussed with the research team with either the most common condition or a representative term used but it is still subjective labelling. Cluster 1 for instance was named as alcohol misuse as this was the most common condition for people identified in the cluster but it could also have been labelled as socioeconomic-driven conditions.

### Recommendations

Identification of patients who are most vulnerable based on clustering of conditions across characteristics such as age, sex and level of deprivation should be used to inform public health strategies including direct primary prevention and interventional clinical services to where they are most needed. There is a need for significant investment in preventative and public health measures and to take action on social determinants of health [33]. The clusters of conditions identified in this study may suggest lifestyle interventions, support groups and mental health interventions in the most deprived areas would be a good strategy to focus on. If not, gaps in health inequalities and differences in multimorbidity prevalence observed may very well continue to widen.

### Conclusions

This paper identified that different sub-population groups with multimorbidity need different interventions to prevent and/or manage multimorbidity. Condition clustering in the multimorbid population is mainly influenced by age and also by sex and area-level socioeconomic deprivation. A third of the youngest age group with multimorbidity have alcohol misuse contributing to their multimorbidity. Almost half of the oldest age group have hypertension and cardiac arrhythmia. When considering the clustering of conditions, it is important to consider the age of the people being studied as well as their sex and level of socio-economic deprivation.

### Supporting information

**S1 Fig. Rate of co-occurrence of pairs of conditions per 1000 people with multimorbidity.**
(TIF)

**S2 Fig. Clustering of Conditions among multimorbid (2+ conditions) patients by sex and level of deprivation (excludes conditions with less than 5% prevalence).**
(TIF)

**S3 Fig. Clustering of Conditions among multimorbid (2+ conditions) patients by age groups (only include conditions with 5%+ prevalence).**
(TIF)

**S4 Fig. Clustering of Conditions among multimorbid (2+ conditions) patients by age and deprivation (excludes conditions with less than 5% prevalence).**
(TIF)

**S1 Table. List of Elixhauser Index conditions, abbreviations, ICD10 codes.**
(PDF)

**S2 Table. Prevalence of the conditions among all patient, patients with multimorbidity and complex multimorbidity.**
(PDF)

**S3 Table. Prevalence of the conditions by characteristics of people with multimorbidity.**
(PDF)

**S4 Table. Multimorbidity Clusters of the Conditions among the whole multimorbid population, sex and deprivation subgroups.**
(PDF)

**S5 Table. Multimorbidity Clusters of the Conditions across age subgroups.**
(PDF)

**S6 Table. Multimorbidity Clusters of the Conditions across age-deprivation subgroups.**
(PDF)

## Acknowledgments

We acknowledge the support of the Health Informatics Centre, University of Dundee for managing and supplying the anonymised data and NHS Tayside and Fife for the original data source. This work uses data provided by patients and collected by the NHS as part of their care and support.

## Author Contributions

**Conceptualization:** Utkarsh Agrawal, Paul Kirk, Colin McCowan.

**Data curation:** Adeniyi Francis Fagbamigbe, Colin McCowan.

**Formal analysis:** Adeniyi Francis Fagbamigbe.

**Funding acquisition:** Colin McCowan.

**Investigation:** Adeniyi Francis Fagbamigbe, Amaya Azcoaga-Lorenzo, Briana MacKerron, Eda Bilici Özyiğit, Daniel C. Alexander, Ashley Akbari, Rhiannon K. Owen, Jane Lyons, Spiros Denaxas, Paul Kirk, Ana Corina Miller, Gill Harper, Carol Dezateux, Anthony Brookes, Sylvia Richardson, Krishnarajah Nirantharakumar, Bruce Guthrie, Lloyd Hughes, Umesh T. Kadam, Kamlesh Khunti, Keith R. Abrams, Colin McCowan.

**Methodology:** Adeniyi Francis Fagbamigbe, Utkarsh Agrawal, Amaya Azcoaga-Lorenzo, Eda Bilici Özyiğit, Colin McCowan.

**Project administration:** Adeniyi Francis Fagbamigbe.

**Resources:** Amaya Azcoaga-Lorenzo, Ronan A. Lyons, Paul Kirk, Ana Corina Miller, Gill Harper, Carol Dezateux, Anthony Brookes, Sylvia Richardson, Krishnarajah Nirantharakumar, Bruce Guthrie, Lloyd Hughes, Umesh T. Kadam, Kamlesh Khunti, Keith R. Abrams.

**Software:** Adeniyi Francis Fagbamigbe, Utkarsh Agrawal, Ashley Akbari, Rhiannon K. Owen, Ronan A. Lyons, Krishnarajah Nirantharakumar, Umesh T. Kadam.

**Supervision:** Daniel C. Alexander, Rhiannon K. Owen, Jane Lyons, Ronan A. Lyons, Spiros Denaxas, Paul Kirk, Anthony Brookes, Sylvia Richardson, Bruce Guthrie, Lloyd Hughes, Kamlesh Khunti, Keith R. Abrams.

**Validation:** Adeniyi Francis Fagbamigbe, Utkarsh Agrawal.

**Visualization:** Adeniyi Francis Fagbamigbe.

**Writing – original draft:** Adeniyi Francis Fagbamigbe, Utkarsh Agrawal, Amaya Azcoaga-Lorenzo, Ana Corina Miller, Gill Harper, Colin McCowan.

**Writing – review & editing:** Adeniyi Francis Fagbamigbe, Utkarsh Agrawal, Amaya Azcoaga-Lorenzo, Briana MacKerron, Eda Bilici Özyiğit, Daniel C. Alexander, Ashley Akbari, Rhiannon K. Owen, Jane Lyons, Ronan A. Lyons, Spiros Denaxas, Paul Kirk, Carol Dezateux, Anthony Brookes, Sylvia Richardson, Krishnarajah Nirantharakumar, Bruce Guthrie, Lloyd Hughes, Umesh T. Kadam, Kamlesh Khunti, Keith R. Abrams, Colin McCowan.

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
