## [Decision Letter · Decision Letter 0]

5 Apr 2023

PONE-D-23-03161Clustering long-term health conditions among 67728 people with multimorbidity using electronic health records in ScotlandPLOS ONE

Dear Dr. Fagbamigbe,

Thank you for submitting your manuscript to PLOS ONE. After careful consideration, we feel that it has merit but does not fully meet PLOS ONE’s publication criteria as it currently stands. Therefore, we invite you to submit a revised version of the manuscript that addresses the points raised during the review process.

We look forward to receiving your revised manuscript.

Kind regards,

Mona Pathak, PhD

Academic Editor

PLOS ONE

- https://www.researchgate.net/publication/325289591_Patterns_of_Multimorbidity_in_Middle-Aged_and_Older_Adults_An_Analysis_of_the_UK_Biobank_Data

In your revision ensure you cite all your sources (including your own works), and quote or rephrase any duplicated text outside the methods section. Further consideration is dependent on these concerns being addressed.

Reviewers' comments:

Reviewer's Responses to Questions

**Comments to the Author**

1. Is the manuscript technically sound, and do the data support the conclusions?

Reviewer #1: Partly

Reviewer #2: Yes

Reviewer #3: Partly

2. Has the statistical analysis been performed appropriately and rigorously? 

Reviewer #1: I Don't Know

Reviewer #2: Yes

Reviewer #3: I Don't Know

3. Have the authors made all data underlying the findings in their manuscript fully available?

Reviewer #1: Yes

Reviewer #2: No

Reviewer #3: No

4. Is the manuscript presented in an intelligible fashion and written in standard English?

Reviewer #1: Yes

Reviewer #2: Yes

Reviewer #3: Yes

5. Review Comments to the Author

Reviewer #1: Clustering long-term health conditions among 67728 people with multimorbidity using electronic health records in Scotland

This paper used 20 years of linked electronic health records to examine clusters of multimorbidity in Scotland across age groups, sex, and levels of deprivation. Looking at how conditions cluster together across different strata in society will potentially have implications for health service delivery. This paper contains important work but presents many complex findings which can make it difficult to interpret at times. Therefore, this paper requires restructuring and some considerable changes before it can be considered for publication.

Abstract

You say multimorbidity is present in 21.3% and then later on you say ‘multimorbidity was present in one of every 5 people’. This is repetition. Consider removing one sentence.

Please reconsider how you describe the multimorbidity clusters in the Abstract as it is confusing in its current form. I would recommend naming the five distinct clusters and then mention that they differ by age, sex, and deprivation without going into too much detail about this in the Abstract.

There are more incidences of repetition in the Abstract – please address this.

Background

The Background is well-structured and you build your argument well.

Please provide a definition of ‘complex multimorbidity’ in order to justify the aims of your study.

Further justification is required for using data from Scotland. For example, the strong socioeconomic gradient in Scotland might be one reason for focusing on this country (example in https://www.nature.com/articles/s41598-022-20546-4#ref-CR18)

Methods

Please clarify – if participants comprised all those who were alive on 31st December 2018, how did you use death records to measure multimorbidity?

It would be helpful to provide justification for the use of the Elixhauser Comorbidity Index to measure multimorbidity when there are more comprehensive measures available now, e.g. https://bmjmedicine.bmj.com/content/bmjmed/1/1/e000247.full.pdf

In the Outcome section, please provide some examples of physical and mental long-term conditions that were included in the multimorbidity definition.

Line 119 – frequencies and proportions of what? Numbers of conditions? This needs to be more clear.

The role of complex multimorbidity in this study is not clear from the Methods- particularly the Data Management and Statistical Analysis section. Please provide more clarity.

Could you include a sentence or two in the Data Management and Statistical Analysis section about Market Basket Analysis (MBA) i.e. how this clustering method allows for clustering across the characteristics of the study sample? I believe hierarchical cluster analysis can do the same thing so it would be good to further justify the use of MBA.

Results

In the Methods you state that the study was based on ‘cross-sectional analysis…when an individual was aged at least 25 years’ (line 90/91). However, the sample in the Results comprise people aged over 44 years. Some clarity is required here. Why over 44 years? Is this because there is a retrospective study period of 20 years?

Were those with complex multimorbidity included in the overall multimorbidity count? i.e. Did 67,728 people have MM or did 87,851 people have MM? Please provide clarity in the paper.

Table 1 – please provide proportions (%) for ‘All individuals’, e.g., for sex, age groups, etc. to allow comparison with the MM group. Please also provide a mean age for ‘All individuals’.

Figure 1 – I would recommend removing the percentages attached to the lines in the graph

In the Introduction, you talk about links between MM and obesity and harmful use of alcohol and drugs (lines 58-60). However, you include both obesity and alcohol abuse as long-term conditions in your multimorbidity measure. I would consider removing this from the Introduction as it introduces confusion.

It might be useful to assign more meaningful names to the clusters, seeing as they don’t differ massively between sex or deprivation levels. For example, cluster 5 could be called the cardiovascular cluster.

Lines 181 to 190 are confusing for the reader. Please consider presenting these results in a different way.

Line 191-192: Please provide some examples of the core set of conditions across strata

Table 2 is really helpful for the interpretation of Results. However, I would suggest reorganising the ‘Clustering’ section of the Results for more clarity. More subheadings would be useful. I would consider basing these on individual clusters.

Why was there no cluster analysis performed amongst those with complex multimorbidity?

Presenting clusters across age groups is difficult and I think Table 3 requires some further clarification. Perhaps include a footnote to indicate that efforts were made to align clusters that were similar. I would also explain why blank cells exist for certain age groups. Moreover, would there be a way of showing how common it is to for participants with MM to belong to one of these clusters in the Table?

Discussion

In the second paragraph, you state that 99% of those aged >80 years were included in cluster 1- please correct to Cluster 2. And correct ‘clusters 2 and 3’ to ‘4 and 6’ (judging by Table 3).

Parts of the Discussion feel like a Results section (but for the Supplementary results). Make sure to discuss results here, rather than just describe them.

Please add some strengths of the study

Are there any limitations relating to market basket analysis that need to be mentioned? Is this widely applied to healthcare data?

Are there any limitations around the measure of deprivation used?

General remarks:

Please check spelling and grammar throughout

Reviewer #2: Thank you for the opportunity to review this paper. An understanding of how LTCs cluster together for people in different social groups (by age, deprivation etc) is very much needed. I was looking for such a paper recently to understand socioeconomic inequalities in outcomes for people with MLTCs so I am pleased to see this work has been done. The paper is clear, well-written and nicely summarises a complex picture. My comments are minor.

Methods:

Please make it clearer when describing the method and also in the tables that a person can "belong" to more than one cluster.

What is the rationale for using Elixhauser conditions?

Results:

S4 Table. I think these important results should be included in the main text (or at least the whole population rows at the top of the table if there isn't room for all of this).

Would it be appropriate to calculate the prevalence of each cluster using the whole sample rather than the number with MLTCs as the denominator? That would be interesting to include in the main text.

S4 Table Least deprived population Solid tumour w/c metastasis cluster appears twice (presumably a labelling error).

S2 Fig. What does the y-axis Height refer to? Please add a footnote to the figure to explain this. Also the clusters are shown as branches and sub-branches. It would be helpful to have a brief explanation in a footnote on what this means for those of us who aren't familiar with the clustering method used.

Discussion:

As the authors point out, the specifics of the clusters identified will depend on which LTCs are counted and whether the data is based on hospital admissions or also primary care. However, it would be helpful to know to what extent the authors think the clusters they have identified are likely to be replicated in broad terms (eg do they make clinical sense, are the results from this local study likely to be generalisable in other places). Or is this more a proof of concept paper showing that we need to segment and not assume that one size fits all?

I think the paper currently makes the latter point well. But if there are some clusters that are likely generalisable then it would be good to expand the policy/practice implications (e.g. for the alcohol, depression, drug use cluster) .

Quite a lot of the material on page 11-12 looks like results rather than discussion. I'd suggest moving some of that up into the results section.

It's not helpful to refer to cluster numbers without a description of the conditions in the discussion.

Some of the combinations have small cell sizes. To what extent is this a limitation here?

Reviewer #3: Title: Clustering long-term health conditions among 67728 people with multimorbidity using electronic health records in Scotland

Manuscript ID: PONE-D-23-03161

In this paper, the authors use secondary data to define a sample of middle-aged and older adults with at least 2 chronic conditions and characterize common condition clusters across age, sex, and socioeconomic deprivation strata. This is an interesting study but there a few issues that I would like to see the authors better address:

- I think it would helpful for the authors to define complex multimorbidity in the Introduction and why it is important to distinguish from multimorbidity more generally.

- Given that we know that certain conditions are going to be more common among older people (or younger people), I think it would be useful to have a stronger argument for this work. Much of the research that I’ve seen that has attempted to understand disease clustering does focus on specific age groups because of this issue. I would like to see the authors be clearer on how this study contributes to the existing research on multimorbidity clustering.

- I found the description of the sample and look back a little bit confusing. I think a figure would help to make this clearer. Also, how was death data used? My understanding is that the sample was everyone alive on Dec. 31st 2018 with a look back to Jan. 1st 2000.

- The diagnoses considered in the analysis are quite mixed. For example, some have argued that diagnoses such as hypertension and obesity are more risk factors rather than chronic conditions in-and-of themselves. I think this kind of issue warrants some discussion.

- Related to the above, common conditions (like arthritis) are what frequently make up the common clusters, often across the different strata considered in this study. This has been reported elsewhere, as well. I think it is worth discussing what this means for thinking of how to design and target services.

- I think that the reliance on hospital and secondary care codes is a more important limitation than described by the authors. I do think that they should address how their estimates compare to those derived from more comprehensive data sources as well as the potential for access to services (even in health systems with universal coverage) to impact on observations across the strata.

- I found the Recommendations to be somewhat of a reach. While I agree that understanding these types of patterns can be used to help identify where services are needed, I also think this should be discussed within the context of currently available and successful prevention and care strategies. I also think that research like this is more directly relevant to informing how we study and understand multimorbidity (rather than specific practice or policy recommendations).

6. PLOS authors have the option to publish the peer review history of their article (what does this mean?). If published, this will include your full peer review and any attached files.

Reviewer #1: **Yes: **Dr Amy Ronaldson

Reviewer #2: No

Reviewer #3: No

---

## [Author Response · Author response to Decision Letter 0]

13 Jun 2023

16th May 2023

Dear 

PONE-D-23-03161

Clustering long-term health conditions among 67728 people with multimorbidity using electronic health records in Scotland

Thank you for your email dated 5th April 2023 asking us for a revision of this paper. 

We are grateful to the reviewers for their thoughtful and constructive feedback, which we have very carefully considered. We provide below a point-by-point response to the points made. We have also included a response to the editorial points.

We have uploaded a version of the revised manuscript with tracked changes and a clean unmarked copy.

Responses to peer-review feedback

Editors Comments

Thank you. We have checked the manuscript against the templates and made the necessary changes.

- https://www.researchgate.net/publication/325289591_Patterns_of_Multimorbidity_in_Middle-Aged_and_Older_Adults_An_Analysis_of_the_UK_Biobank_Data

In your revision ensure you cite all your sources (including your own works), and quote or rephrase any duplicated text outside the methods section. Further consideration is dependent on these concerns being addressed.

Thank you for highlighting this may simply be due to a common use of familiar language. We have read both papers alongside each other and could not see any major overlap so with the changes we have included hope this has been resolved. If there are specific areas, you would still wish us to rewrite we would be happy to address these.

We have updated the Data Sharing statement at the end of the manuscript to reflect the data are not publicly available but could be accessed via application and granting of subsequent approvals.

Page 17 Paragraph 3

“The data used in this study are sensitive and are not publicly available. Access to the data is by application to the Health Informatics Centre, University of Dundee, Scotland (hicsupport@dundee.ac.uk) using their standard governance and access processes.”

Thank you. We have changed the captions and updated citations as requested.

Reviewer #1: Clustering long-term health conditions among 67728 people with multimorbidity using electronic health records in Scotland

R1.1- This paper used 20 years of linked electronic health records to examine clusters of multimorbidity in Scotland across age groups, sex, and levels of deprivation. Looking at how conditions cluster together across different strata in society will potentially have implications for health service delivery. This paper contains important work but presents many complex findings which can make it difficult to interpret at times. Therefore, this paper requires restructuring and some considerable changes before it can be considered for publication.

Thank you. We have incorporated all the reviewers’ comments and restructured a number of sections

Abstract

R1.2- You say multimorbidity is present in 21.3% and then later on you say ‘multimorbidity was present in one of every 5 people’. This is repetition. Consider removing one sentence.

Thank you. We have removed the latter statement at line 64.

R1.3- Please reconsider how you describe the multimorbidity clusters in the Abstract as it is confusing in its current form. I would recommend naming the five distinct clusters and then mention that they differ by age, sex, and deprivation without going into too much detail about this in the Abstract.

We have named the five clusters seen in the overall population as requested and shortened and amended the abstract as below.

“We identified five distinct clusters of conditions in the population with multimorbidity: alcohol misuse, cancer, obesity, renal failure and heart failure. Clusters of long-term conditions differed by age, sex and socioeconomic deprivation, with some clusters not present for specific strata and others including additional conditions.”

R1.4- There are more incidences of repetition in the Abstract – please address this.

We have reviewed the abstract and removed repetition where identified.

Background

The Background is well-structured and you build your argument well.

We thank the reviewer for their positive comments

R1.5- Please provide a definition of ‘complex multimorbidity’ in order to justify the aims of your study.

We have changed the text to define this and added a reference to previous work where this definition has been used (line 94).

“and complex multimorbidity (defined as having 4 or more multiple long-term conditions)(Agrawal et al., 2022)333(Agrawal et al., 2022)333”

R1.6- Further justification is required for using data from Scotland. For example, the strong socioeconomic gradient in Scotland might be one reason for focusing on this country (example in https://www.nature.com/articles/s41598-022-20546-4#ref-CR18)

We have now explained why dataset was generated and how the need for this work was decided (Lines 113-116).

“The data was generated for a study exploring multimorbidity across different nations in the UK. Exploration of the data showed a strong socioeconomic and age gradient in terms of individual conditions and multimorbidity which we felt warranted further exploration.(Cezard et al., 2022)202020(Cezard et al., 2022)202020”. Scotland is one of the UK nations and was purposively chosen for this study because of its strong socioeconomic gradient.

Methods

R1.7- Please clarify – if participants comprised all those who were alive on 31st December 2018, how did you use death records to measure multimorbidity?

We did have access to the Deaths data but as stated by the reviewer this was not used to identify multimorbidity for patients alive at 31st December 2018. We have changed text as below (Lines 117-119, and Lines 131-132).

“Within this study, we focussed on individuals who were alive on 31st December 2018 with dates of death ascertained from linked National Record Scotland death certificates and the population register.”

“All hospital admissions, psychiatric hospital admissions, outpatients, cancer registry and emergency department records over the period were examined, and all the International Classification of Diseases (ICD)-10 codes were extracted.”

R1.8- It would be helpful to provide justification for the use of the Elixhauser Comorbidity Index to measure multimorbidity when there are more comprehensive measures available now, e.g. https://bmjmedicine.bmj.com/content/bmjmed/1/1/e000247.full.pdf

The Ho index or the code list suggested by the reviewer was not available at the time of this work for its use against EHR. We chose the ELixhauser as it has been shown to be a good marker of mortality and is well established for use with EHR. We have added text to reflect this as requested (Lines 137-139)

“The Elixhauser Index was chosen as previous reviews have suggested it is a good marker for mortality, although this was not an outcome of interest in this study, and it is well established for use with electronic health record and used in earlier studies [27]“ 

 R1.9- In the Outcome section, please provide some examples of physical and mental long-term conditions that were included in the multimorbidity definition.

We have included a comprehensive list in the Supplementary S1 Table and added to the text (Lines 136-137).

“Depression and psychoses are examples of mental health long-term conditions included within the Elixhauser index whilst weight loss and cancer are some of the physical conditions listed (see Table S1 for full list of conditions and related codes).”

R1.10- Line 119 – frequencies and proportions of what? Numbers of conditions? This needs to be more clear.

We have changed this to specify this relates to numbers of individuals with the condition (Line 150).

“Frequencies and proportions of individuals with the conditions within each stratum”

R1.11- The role of complex multimorbidity in this study is not clear from the Methods- particularly the Data Management and Statistical Analysis section. Please provide more clarity.

We reported on complex multimorbidity in Table 1 as we felt that it was important to show the complexity that certain people with multimorbidity have and therefore the need to look in more depth at clusters of conditions. The majority of cluster identified for the whole population but especially within strata had more than 2 conditions present. We believe it is an important descriptive statistic to report but it did not need further exploration. We have added text to show this as below (Lines 151-152).

“We reported the prevalence of multimorbidity (2+) and complex multimorbidity (4+) to show how this changed across strata of the population.”

R1.12- Could you include a sentence or two in the Data Management and Statistical Analysis section about Market Basket Analysis (MBA) i.e. how this clustering method allows for clustering across the characteristics of the study sample? I believe hierarchical cluster analysis can do the same thing so it would be good to further justify the use of MBA.

We have added to the text to justify the selection of Market Basket Analysis as our clustering algorithm.

“We used Market Basket Analysis (MBA) because as it has been reported as more efficient for binary (present/absent) outcomes than the hierarchical cluster analysis that was originally built for quantitative outcomes (Hahsler et al., 2005; Hahsler & Karpienko, 2017)(Hahsler et al., 2005; Hahsler & Karpienko, 2017). It also allows an individual to "belong" to more than one cluster if they have a large number of different conditions.”

Results

R1.13- In the Methods you state that the study was based on ‘cross-sectional analysis…when an individual was aged at least 25 years’ (line 90/91). However, the sample in the Results comprise people aged over 44 years. Some clarity is required here. Why over 44 years? Is this because there is a retrospective study period of 20 years?

We apologise for the confusion. We have amended the first sentence of the Methods to better describe the population for the study (Lines110-112).

“The population for this study were residents of Fife and Tayside, Scotland who were aged at least 25 years old on 1st January 2000 and who were followed up until 31 December 2018, when a cross-sectional analysis of all live patients was performed.”

R1.14- Were those with complex multimorbidity included in the overall multimorbidity count? i.e. Did 67,728 people have MM or did 87,851 people have MM? Please provide clarity in the paper.

Those with complex multimorbidity were also included in the multimorbidity count and we have therefore amended the text to better reflect this (Lines 182-184).

“Overall, 318,235 people aged 44 years and over were included in the analysis, with 67,728(21·3%) having multimorbidity, while 20,123(6·3%) were also classed as having complex multimorbidity.”

R1.15- Table 1 – please provide proportions (%) for ‘All individuals’, e.g., for sex, age groups, etc. to allow comparison with the MM group. Please also provide a mean age for ‘All individuals’.

We have provided this in Table 1 and added a mean age for the overall population, those with multimorbidity and those with complex multimorbidity.

R1.16- Figure 1 – I would recommend removing the percentages attached to the lines in the graph

We have removed the percentages in the Figure as requested.

R1.17- In the Introduction, you talk about links between MM and obesity and harmful use of alcohol and drugs (lines 58-60). However, you include both obesity and alcohol abuse as long-term conditions in your multimorbidity measure. I would consider removing this from the Introduction as it introduces confusion.

We have removed this as suggested (Line 79).

R1.18- It might be useful to assign more meaningful names to the clusters, seeing as they don’t differ massively between sex or deprivation levels. For example, cluster 5 could be called the cardiovascular cluster.

Thank you, we have assigned names to the clusters. We have Alcohol misuse Cluster, Cancer Cluster, Obesity Cluster, Renal Failure Cluster and Heart Failure Cluster

R1.19- Lines 181 to 190 are confusing for the reader. Please consider presenting these results in a different way.

We hare rewritten this section in L306-315

A number of previous studies have identified clusters that include cardiovascular-metabolic conditions, mental health issues, and musculoskeletal disorders (Prados-Torres et al., 2014). A recent study has also demonstrated variations in clusters by age, as well as differences in mortality and service utilisation(Zhu et al., 2020). In addition, our study has found how long-term conditions cluster differently based on age, sex, and socioeconomic deprivation:

R1.20- Line 191-192: Please provide some examples of the core set of conditions across strata.

We have provided the core sets (Lines 246-247).

“The core conditions identified across strata included alcohol misuse, other neurological disorders and depression in the alcohol ,misuse cluster and solid tumour without metastasis and metastatic cancer in the cancer cluster.”

R1.21- Table 2 is really helpful for the interpretation of Results. However, I would suggest reorganising the ‘Clustering’ section of the Results for more clarity. More subheadings would be useful. I would consider basing these on individual clusters.

We have Introduced subheadings as requested in Table 2.

R1.22- Why was there no cluster analysis performed amongst those with complex multimorbidity?

We have ammended the text as explained above to establish that complex multimorbidity and how it differs across strata was part of the rationale for exploring clusters. We do not feel that clustering conditions purely for those individuasl with complex multimorbidity would add to this paper.

R1.23- Presenting clusters across age groups is difficult and I think Table 3 requires some further clarification. Perhaps include a footnote to indicate that efforts were made to align clusters that were similar. I would also explain why blank cells exist for certain age groups. Moreover, would there be a way of showing how common it is to for participants with MM to belong to one of these clusters in the Table?

Thank you. We have provided the footnotes. Yes, we presented the how common it is to for participants with MM to belong to one of these clusters in S4 and S5 Table

Discussion

R1.24- In the second paragraph, you state that 99% of those aged >80 years were included in cluster 1- please correct to Cluster 2. And correct ‘clusters 2 and 3’ to ‘4 and 6’ (judging by Table 3).

Thank you. This was based on Supplementary S5 Table. We have now harmonized the numbering in S5 Table with Table 3. We have made this clearer in the text.

R1.25- Parts of the Discussion feel like a Results section (but for the Supplementary results). Make sure to discuss results here, rather than just describe them.

Thank you. We have revised the entire discussion section

R1.26- Please add some strengths of the study

We have added strengths of the study (Page 18).

“The subjects for the study were drawn from the population within two Scottish regions with comprehensive health records over a long period of time. The use of well-defined conditions and ICD-10 codesets to identify each condition allows other researchers to explore multimorbidity using the same methods. Using market basket analysis to cluster conditions rather than classifying patients into mutually exclusive groups likely represents real-world scenarios more accurately. Patients with multiple long-term conditions can be classified in different ways depending on the specific factors that are taken into consideration. Using the same approach across different strata meant comparisons were down to the underlying data rather than simply different populations using different methods.”

R1.27- Are there any limitations relating to market basket analysis that need to be mentioned? Is this widely applied to healthcare data?

We have added a possible limitation of measure of deprivation. The method has been applied to several healthcare data. However, the clusters generated by market basket analysis are based on patterns in the data, and may not guarantee association, so must be treated with caution. Page 18

“The measure of deprivation used in this study is allocated at a postcode level but it a small area approximation rather than a direct measure of individual deprivation.”

R1.28- Are there any limitations around the measure of deprivation used?

And similarly for Market basket analysis.

“”Similarly there have been a number of different methods used to identify clusters within multimorbid populations but our choice of Market Basket Analysis as a methodological tool is unlikely to be the cause of differences when examining strata.” Page 18

General remarks:

R1.29- Please check spelling and grammar throughout 

Thank you, we have carried out language edit

Reviewer #2: Thank you for the opportunity to review this paper. An understanding of how LTCs cluster together for people in different social groups (by age, deprivation etc) is very much needed. I was looking for such a paper recently to understand socioeconomic inequalities in outcomes for people with MLTCs so I am pleased to see this work has been done. The paper is clear, well-written and nicely summarises a complex picture. My comments are minor.

Thank you for your kind comments.

Methods:

R2.1- Please make it clearer when describing the method and also in the tables that a person can "belong" to more than one cluster.

Thank you, we have made this clearer in Lines 168-171 as reported to Reviewer 1.

R2.2- What is the rationale for using Elixhauser conditions?

Thank you, we have added a rationale for this in Lines 137-139 as reported to Reviewer 1.

Results:

R2.3- S4 Table. I think these important results should be included in the main text (or at least the whole population rows at the top of the table if there isn't room for all of this).

Thank you. We have created Table 4 to reflect this.

R2.4- Would it be appropriate to calculate the prevalence of each cluster using the whole sample rather than the number with MLTCs as the denominator? That would be interesting to include in the main text.

In S4 Table, we presented the % of MM population clustered within each sub-population group. We do not think it is appropriate to compute prevalence for the general population that didn’t have multimorbidity

R2.5- S4 Table Least deprived population Solid tumour w/c metastasis cluster appears twice (presumably a labelling error).

Thank you for highlighting this error. We have corrected this in the Table

R2.6- S2 Fig. What does the y-axis Height refer to? Please add a footnote to the figure to explain this. Also the clusters are shown as branches and sub-branches. It would be helpful to have a brief explanation in a footnote on what this means for those of us who aren't familiar with the clustering method used.

Thank you. We have added footnotes to explain this.

Discussion:

R2.7- As the authors point out, the specifics of the clusters identified will depend on which LTCs are counted and whether the data is based on hospital admissions or also primary care. However, it would be helpful to know to what extent the authors think the clusters they have identified are likely to be replicated in broad terms (eg do they make clinical sense, are the results from this local study likely to be generalisable in other places). Or is this more a proof of concept paper showing that we need to segment and not assume that one size fits all?

I think the paper currently makes the latter point well. But if there are some clusters that are likely generalisable then it would be good to expand the policy/practice implications (e.g. for the alcohol, depression, drug use cluster) .

We have added the text below in response to the reviewer’s comment. L415-431.

The identified clusters make clinical sense and are likely to be generalisable to similar populations, but many of the implications they raise for clinical services are known. Related or consequential conditions have separate codes and hence induce clustering which is somewhat artificial and commonly represents different manifestations of disease pathways or disease progression. The benefit of this work is identifying that stratifying by age, sex and socioeconomic status is needed to identify the most relevant clusters for those groups. 

For instance, alcohol cluster included alcohol abuse, depression and other neurological disorders. This is not surprising as alcoholism may lead to depression or coping with depression through excess drinking. Alcohol is a known neurotoxin and hence a relationship with other neurological conditions is to be expected. Alcohol raises blood pressure and hence stroke risk. The cancer cluster includes tumour without metastasis and metastatic cancer, this is expected as cancers progress. For the obesity cluster, obesity is a risk factor for hypertension and diabetes which will be a common group in this cluster. Hypothyroidism is more common in rheumatoid arthritis. In the renal failure cluster, atherosclerosis is a common cause of renal failure and peripheral vascular disorders. Iron deficiency anaemia is very common in renal failure and a consequence of erythropoietin deficiency. Also, fluid & electrolyte disorders are a consequence of renal failure. For the heart failure cluster, atherosclerosis or valvular disorders can cause heart failure and arrythmias.

R2.8- Quite a lot of the material on page 11-12 looks like results rather than discussion. I'd suggest moving some of that up into the results section.

Thank you. We have changed the discussion to reflect this point which was also raised by reviewer 1. See above for details. Page 11-12

R2.9- It's not helpful to refer to cluster numbers without a description of the conditions in the discussion.

We have named the clusters so that referring to them is easier and to try and describe something meaningful about each cluster. Page 9-10, Tables 2 and 3

R2.10- Some of the combinations have small cell sizes. To what extent is this a limitation here?

The small cell counts do not constitute a limitation as we didn’t encounter any issue in our algorithms while implementing the clustering.

Reviewer #3: Title: Clustering long-term health conditions among 67728 people with multimorbidity using electronic health records in Scotland

Manuscript ID: PONE-D-23-03161

In this paper, the authors use secondary data to define a sample of middle-aged and older adults with at least 2 chronic conditions and characterize common condition clusters across age, sex, and socioeconomic deprivation strata. This is an interesting study but there a few issues that I would like to see the authors better address:

Thank you

R3.1- I think it would helpful for the authors to define complex multimorbidity in the Introduction and why it is important to distinguish from multimorbidity more generally.

We have addressed this. L102, page 3

R3.2- Given that we know that certain conditions are going to be more common among older people (or younger people), I think it would be useful to have a stronger argument for this work. Much of the research that I’ve seen that has attempted to understand disease clustering does focus on specific age groups because of this issue. I would like to see the authors be clearer on how this study contributes to the existing research on multimorbidity clustering.

Besides the clustering of the conditions among all the MM participants, we conducted age-specific, sex-specific and deprivation-specific clustering. Kindly see Tables 2 and 3 and the supplementary materials. This is the strength of this study, and it is addition to the body of knowledge.

R3.3- overall clu

- I found the description of the sample and look back a little bit confusing. I think a figure would help to make this clearer. Also, how was death data used? My understanding is that the sample was everyone alive on Dec. 31st 2018 with a look back to Jan. 1st 2000.

We have changed the text to better explain how death data was used in identifying the cohort and also to explain how the cohort was formed and the follow-up period identified.- See response to Reviewer 1. Page 4

R3.4- The diagnoses considered in the analysis are quite mixed. For example, some have argued that diagnoses such as hypertension and obesity are more risk factors rather than chronic conditions in-and-of themselves. I think this kind of issue warrants some discussion.

We agree that hypertension and obesity can be either depending on the context but the conditions we investigated were all conditions identified within the Elixhauser conditions and we discuss the limitations of this approach in the Discussion (Page 16). In the current study, we only assessed clustering of conditions, we didn’t assess risk factors of any outcome. 

R3.5- Related to the above, common conditions (like arthritis) are what frequently make up the common clusters, often across the different strata considered in this study. This has been reported elsewhere, as well. . 

We do agree, we have provided possible linkages among the conditions in the clusters L415-432.

R3.6- I think that the reliance on hospital and secondary care codes is a more important limitation than described by the authors. I do think that they should address how their estimates compare to those derived from more comprehensive data sources as well as the potential for access to services (even in health systems with universal coverage) to impact on observations across the strata.

We agree. We have rewritten the limitation section to reflect this Page 18

R3.7- I found the Recommendations to be somewhat of a reach. While I agree that understanding these types of patterns can be used to help identify where services are needed, I also think this should be discussed within the context of currently available and successful prevention and care strategies. I also think that research like this is more directly relevant to informing how we study and understand multimorbidity (rather than specific practice or policy recommendations).

Thank you. We have re-focussed and contextualized the recommendations. P477-484

---

## [Decision Letter · Decision Letter 1]

24 Oct 2023

PONE-D-23-03161R1Clustering long-term health conditions among 67728 people with multimorbidity using electronic health records in ScotlandPLOS ONE

Dear Dr. Fagbamigbe,

Thank you for submitting your manuscript to PLOS ONE. After careful consideration, we feel that it has merit but does not fully meet PLOS ONE’s publication criteria as it currently stands. Therefore, we invite you to submit a revised version of the manuscript that addresses the points raised during the review process.

We look forward to receiving your revised manuscript.

Kind regards,

Sreeram V. Ramagopalan

Academic Editor

PLOS ONE

Journal Requirements:

Reviewers' comments:

Reviewer's Responses to Questions

**Comments to the Author**

1. If the authors have adequately addressed your comments raised in a previous round of review and you feel that this manuscript is now acceptable for publication, you may indicate that here to bypass the “Comments to the Author” section, enter your conflict of interest statement in the “Confidential to Editor” section, and submit your "Accept" recommendation.

Reviewer #2: All comments have been addressed

Reviewer #3: (No Response)

2. Is the manuscript technically sound, and do the data support the conclusions?

Reviewer #2: Yes

Reviewer #3: Partly

3. Has the statistical analysis been performed appropriately and rigorously? 

Reviewer #2: Yes

Reviewer #3: N/A

4. Have the authors made all data underlying the findings in their manuscript fully available?

Reviewer #2: No

Reviewer #3: Yes

5. Is the manuscript presented in an intelligible fashion and written in standard English?

Reviewer #2: Yes

Reviewer #3: Yes

6. Review Comments to the Author

Reviewer #2: (No Response)

Reviewer #3: Title: Clustering long-term health conditions among 67728 people with multimorbidity using electronic health records in Scotland

Manuscript ID: PONE-D-23-03161R

The authors have clearly put in quite a bit of effort to address the Reviewer’s concerns. I appreciate that they have clarified their methods and presentation of their results. There are still a few issues that I would like to see addressed:

1. It is not clear to me why Cluster 1 is characterized as Alcohol Abuse. I would like to see some rationale for this decision. For the other clusters, the reasoning for the label is more clear or seems to be more representative of the main conditions but less so for Cluster 1. I think because a label like alcohol abuse could be considered stigmatizing, and the way it is presented with economic deprivation, makes me somewhat uncomfortable. I also wonder if there is something to be discussed in terms of how codes such as alcohol abuse are assigned and whether that is at all related to the observed patterns with age and economic deprivation.

2. I think that the Discussion would benefit from some more thorough editing. There are some places where the sentences are not complete – I think this is likely from the multiple rounds of track changes – but it does make it difficult to follow even the clean version. It is also repetitive in places, particularly the points about clustering showing variability by age and economic deprivation.

3. Related to the above, I think that the authors can provide a deeper discussion of what the clusters might say about why they may be seeing these conditions together – either because they are physiologically related, have common determinants, or some other reason. I do not think as currently presented there is much offered beyond describing the common conditions in clusters.

4. I do not understand the statement that Market Basket Analysis is based on patterns in the data but may not guarantee associations (page 15, line 372).

7. PLOS authors have the option to publish the peer review history of their article (what does this mean?). If published, this will include your full peer review and any attached files.

Reviewer #2: No

Reviewer #3: No

---

## [Author Response · Author response to Decision Letter 1]

30 Oct 2023

Comments to the Author

Reviewer #2: (No Response)

Reviewer #3: Title: Clustering long-term health conditions among 67728 people with multimorbidity using electronic health records in Scotland

Manuscript ID: PONE-D-23-03161R

The authors have clearly put in quite a bit of effort to address the Reviewer’s concerns. I appreciate that they have clarified their methods and presentation of their results. There are still a few issues that I would like to see addressed:

1. It is not clear to me why Cluster 1 is characterized as Alcohol Abuse. I would like to see some rationale for this decision. For the other clusters, the reasoning for the label is more clear or seems to be more representative of the main conditions but less so for Cluster 1. I think because a label like alcohol abuse could be considered stigmatizing, and the way it is presented with economic deprivation, makes me somewhat uncomfortable. I also wonder if there is something to be discussed in terms of how codes such as alcohol abuse are assigned and whether that is at all related to the observed patterns with age and economic deprivation.

The naming of the clusters was discussed with the research team with either the most common condition or a representative term used but it is still subjective labelling. Cluster 1 for instance was named as alcohol misuse as this was the most common condition for people identified in the cluster but it could also have been labelled as socioeconomic-driven conditions

2. I think that the Discussion would benefit from some more thorough editing. There are some places where the sentences are not complete – I think this is likely from the multiple rounds of track changes – but it does make it difficult to follow even the clean version. It is also repetitive in places, particularly the points about clustering showing variability by age and economic deprivation.

We have edited the discussion and removed duplications

3. Related to the above, I think that the authors can provide a deeper discussion of what the clusters might say about why they may be seeing these conditions together – either because they are physiologically related, have common determinants, or some other reason. I do not think as currently presented there is much offered beyond describing the common conditions in clusters.

We have edited the section as well. Our analysis shows association not causality, but it may be possible to surmise the drivers of specific clusters as identified. Conditions most prevalent in our most deprived population groups include alcohol and drug misuse, depression and obesity which are all know to be associated with social factors. Other identified clusters are likely to have more physiological drivers e.g. hypertension through to heart diseases.

4. I do not understand the statement that Market Basket Analysis is based on patterns in the data but may not guarantee associations (page 15, line 372).

Thank you, We have changed this line to now read “However, the clusters generated by market basket analysis are based on empirical patterns in the data and only show associations between different conditions, it does not show any causal relationships between those data.”

---

## [Editor Report · Decision Letter 2]

7 Nov 2023

Clustering long-term health conditions among 67728 people with multimorbidity using electronic health records in Scotland

PONE-D-23-03161R2

Dear Dr. Fagbamigbe,

We’re pleased to inform you that your manuscript has been judged scientifically suitable for publication and will be formally accepted for publication once it meets all outstanding technical requirements.

Kind regards,

Sreeram V. Ramagopalan

Academic Editor

PLOS ONE
---

## [Editor Report · Acceptance letter]

17 Nov 2023

PONE-D-23-03161R2 

Clustering long-term health conditions among 67728 people with multimorbidity using electronic health records in Scotland 

Dear Dr. Fagbamigbe:

I'm pleased to inform you that your manuscript has been deemed suitable for publication in PLOS ONE. Congratulations! Your manuscript is now with our production department. 

Kind regards, 

on behalf of

Dr. Sreeram V. Ramagopalan 

Academic Editor

PLOS ONE